# Correlations of the Osteoporosis Self-Assessment Tool for Asians (OSTA) and Three Panoramic Indices Using Quantitative Ultrasound (QUS) Bone Densitometry

**DOI:** 10.3390/dj11020034

**Published:** 2023-01-30

**Authors:** Bramma Kiswanjaya, Hanna H. Bachtiar-Iskandar, Akihiro Yoshihara

**Affiliations:** 1Department of Dentomaxillofacial Radiology, Faculty of Dentistry, Universitas Indonesia, Jl. Salemba Raya No. 4, Jakarta 10430, Indonesia; 2Division of Oral Science for Health Promotion, Department of Oral Health Science and Promotion, Graduate School of Medical and Dental Sciences, Niigata University, Niigata-City 951-8514, Japan

**Keywords:** older adults, osteoporosis, panoramic radiography, quantitative ultrasound

## Abstract

This study aimed to evaluate the correlation between the Osteoporosis Self-Assessment Tool for Asians (OSTA) and three panoramic indices in relation to z-score and t-score values using quantitative ultrasound (QUS) bone densitometry. The sensitivity, specificity, and area under the curve (AUC) of the OSTA index were also measured using the QUS tool to evaluate the method’s performance in identifying people at risk of osteoporosis. The study employed a cross-sectional design with 387 participants (190 men, 197 women). Patients’ mandibular cortical indexes (MCI), mandibular cortical widths (MCW), and panoramic mandibular indexes (PMI) were measured from panoramic images. The sensitivity, specificity, and AUC were calculated using an OSTA index cutoff of ≤−1 and a t-score of ≤−1.0 for the QUS bone densitometry. The coefficient correlation of the OSTA index with the z-score (r = −0.563, *p* < 0.001) and t-score (r = −0.740, *p* < 0.001) shows a higher value than the MCI, MCW, and PMI, per the QUS. The sensitivity, specificity, and AUC values with a cutoff t-score of ≤−1.0 per the QUS in men was 90%, 50%, and 0.812, and in women, 96.8%, 30%, and 0.862. The OSTA index is a simple method that can be used in general dental practice.

## 1. Introduction

Osteoporosis can affect patients’ oral health and dental treatments if the condition is not detected early. Numerous studies have been conducted to determine the relationship between the severity of periodontal disease, tooth loss, and jawbone atrophy in older people at risk of osteoporosis [1,2]. Periodontal disease is closely related to the ability to maintain dental and oral health, and it can, in turn, lead to inflammation and bone remodeling in the pathogenesis of osteoporosis and periodontitis [1]. This condition increases the risk of tooth loss and has an impact on the jawbone’s density, width, and height [2]. In older patients with partial or complete edentulous, the atrophy of the jawbone varies between individuals. Furthermore, the design of prostheses in older patients at risk of osteoporosis requires more careful selective treatments related to the condition and severity of jawbone atrophy [3]. Loss of teeth and ill-fitting prostheses will ultimately affect the function of and satisfaction with mastication and lower the quality of life for those at risk for osteoporosis. Another consideration is that undetected osteoporosis also increases the risk of fractures; such injuries can cause older patients to become bedridden and unproductive, possibly even leading to death [4]. Therefore, finding a quick approach to recognizing patients who are at risk for osteoporosis in regular dentistry practice is crucial so that these patients can receive the best care possible.

Body weight, patient age, and panoramic radiography are typically measured during dentistry appointments, especially for older patients who want to have prostheses made [5]. The Osteoporosis Self-Assessment Tool for Asians (OSTA) is an easy-to-use tool that identifies postmenopausal women at risk of osteoporosis based on patient weight and age [6]. Studies with female participants have been carried out using OSTA and bone mineral density (BMD) measurements from the calcaneus, measured by quantitative ultrasound (QUS) densitometry. Due to the cutoff value of the t-score and limited sample sizes, conflicts in previous findings have been reported [7,8]. Conversely, studies involving male participants have concluded that the performance of the OSTA index does not show the outcome of the QUS index in men, despite one study attempting a change of the cutoff value [9].

Panoramic radiography, in contrast, is an examination frequently used to evaluate the condition of the teeth, periodontal tissue, and surrounding anatomy. It illuminates incidental findings that are not visible during a clinical examination. Furthermore, early osteoporosis patient detection via panoramic images is possible. The mandibular cortical index (MCI) introduced by Klemetti et al. in 1994 [10], mandibular cortical width (MCW) developed by Taguchi et al. in 1996 [11], and panoramic mandibular index (PMI) designed by Benson et al. in 1991 [12] are a few examples of these imaging methods. Although several studies have concluded that OSTA is associated with postmenopausal female patients at risk of osteoporosis, the correlation between OSTA and the method index for detecting osteoporosis from panoramic images has yet to be studied in depth. Another phenomenon needing investigation is the relationship of the value of BMD (stiffness index, or SI), the speed of sound (SOS), and the calcaneus broadband attenuation of sound (BUA) with the QUS tool. In addition to the z-score and t-score, these parameters are typically used as research variables for measuring bone densitometry [9].

The present study drew from a large number of participants, including not only postmenopausal women but also, more generally, women and men aged 50 to 70 years. The indications for dual-energy X-ray absorptiometry (DEXA) are all women over 65 years and men over 70 years, or people aged over 50 years who have experienced a fracture with minimal or no trauma, excluding pathological fractures [13]. The DEXA test is therefore not relevant to this investigation. The purpose of this study was to evaluate the correlation of the OSTA with the three described panoramic indices in relation to z-score and t-score values using QUS bone densitometry. Additionally, the QUS tool was used to test the OSTA index’s sensitivity, specificity, and area under the curve (AUC) in order to assess how well it performed in identifying older individuals who were at risk for osteoporosis. The null hypothesis was that the OSTA index has no correlation with the three panoramic indices in relation to z-score and t-score values using QUS bone densitometry.

## 2. Materials and Methods

Using a cross-sectional design, this study was carried out with an initial 400 healthy older (aged 50 to 70 years) participants from community health centers in the province of West Java, Indonesia, in 2019. The exclusion criteria were patients with a history of fracture or osteoporosis treatment who could not attend all of the examinations independently. Due to anterior and posterior errors or horizontal or vertical errors, 29 radiographs did not pass in the selection of panoramic radiographs. These errors occurred due to patients having difficulty positioning their head and biting the bite tab. Height could not be measured in four subjects because of a bent spine causing difficulty with standing straight. Ultimately, the study included 387 participants, of whom 190 were men and 197 were women. The Dentistry Research Ethics Commission’s research protocol was accepted by the Faculty of Dentistry, Universitas Indonesia (Letter No. 35, Ethical Approval /FKGUI/III/2019), and each participant gave their informed consent.

### 2.1. Osteoporosis Self-Assessment Tool for Asians (OSTA)

An OSTA score is calculated by deducting the patient’s age from their weight and multiplying the result by 0.2, which is represented by:OSTA = 0.2 [weight (kg) − age (year)](1)

Although OSTA is intended to measure women at risk of osteoporosis, in this study, male subjects were also evaluated. The OSTA values are divided into three groups: low risk (index > −1), intermediate risk (index −1 to −4), and high risk (index < −4) [6].

### 2.2. Measurement of the Three Panoramic Indices

The three panoramic indices—the MCI, MCW, and PMI—were measured in all participants on both sides. Veraviewepocs 2D and i-Dixel imaging software (J. Morita Corp., Kyoto, Japan) were used to create digital panoramic radiographs, using a 10 milliampere-seconds (mAs) setting for 12 to 15 s at 70 to 80 kVp. MCI, MCW, and PMI values were measured by specialists with more than 10 years of expertise in oral radiology. Prior to taking the measurements, the image magnification was set to 1, and the radiograph’s brightness and contrast settings were appropriately adjusted to assess the area of interest. To prevent weariness and mistakes in the analysis, interpretation of the MIC, MCW, and PMI values was restricted to 5 to 8 panoramic radiographs per day. The MCI was obtained using the Klemetti index, introduced in 1994 [Figure 1] [10], while the mean MCW was calculated following Taguchi et al. [11]. The mean PMI ratio was measured [Figure 2] according to Benson et al. [12].

Intra- and inter-observer agreement measurements were performed by the two radiologists on 50 randomly selected panoramic images with a seven-day interval between measurements. The MCI was used to generate categorical data in accordance with kappa agreement in order to assess the agreement between intra-observer and inter-observer measurements. The MCW was selected to provide continuous data using a Bland–Altman plot. The MCI’s intra-observer and inter-observer kappa agreements were 0.88 and 0.847, respectively. According to the Bland–Altman test of agreement between the intra-observer and inter-observer for the MCW, the mean difference and standard deviation were 0.02 ± 0.09 and 0.03 ± 0.29, respectively. Following the Bland–Altman test, there was no discernible difference in the measurements of the intra-observer and the inter-observer (*p* > 0.05, one-sample *t*-test). The measurements were taken to be consistent and were applied equally.

### 2.3. Bone Densitometry, Body Mass Index, and Number of Remaining Teeth

QUS bone densitometry was used to measure the values of the z-score and t-score to identify bone density. Individual calcaneal QUS was performed with a Lunar Achilles Insight bone densitometer (GE Healthcare, Milwaukee, WI, USA). The values of the z-scores and t-scores were categorized according to World Health Organization (WHO) criteria [14]. In addition, SI, SOS, and BUA were measured to evaluate their correlation with OSTA. The patient’s weight and height were measured with the same instrument using a medical mechanical body weighing scale (SH-8024, Jiangsu Suhong Medical Device Co., Ltd., Changzhou, China). Weight was measured in kilograms, while height was measured in centimeters. Body mass index (BMI) was obtained by calculating the weight (kg) divided by the square of the height (m). In addition, the number of remaining teeth was analyzed up to the third molars on panoramic radiographs using the following criteria: healthy, caries, or restored. Teeth indicated for extraction and remaining roots were not included.

### 2.4. Statistical Analyses

Initially, the Kolmogorov–Smirnov test was used to analyze the data distribution. The relationships between OSTA and all the variables in men and women were assessed using a one-way ANOVA test and a Kruskal–Wallis test, when applicable. Additionally, when applicable, categorical data were measured using a chi-square test or Fisher’s exact test. A Spearman’s rho correlation coefficient value test was performed to evaluate the correlation value among all the variables based on their z-scores and t-scores. Using QUS as a reference, a receiver operating curve (ROC) was created for the OSTA index. The sensitivity, specificity, and AUC values were calculated using a cutoff of OSTA index ≤ −1 and a t-score of ≤−1.0 in QUS bone densitometry. The power analysis was set at 0.8, and all participants were divided into groups by sex. The data were analyzed using SPSS (version 17.0; IBM Corp., Armonk, NY, USA), with *p* < 0.05 chosen as the level of statistical significance.

## 3. Results

The subjects are described in Table 1. Female participants were, on average, significantly older than the male participants. In terms of QUS, the values of SOS, BUA, and SI were significantly lower in women than men, with a mean t-score in the categories of osteopenia (−2.5 < t-score < −1) for both men and women. The methods used to identify the risk of osteoporosis were OSTA, MCI, and MCW, which differed significantly between men and women. PMI values, in contrast, did not vary significantly. The highest percentages among the male participants’ MCI and OSTA values were in the normal eroded cortex (70.6%) with a lower risk of osteoporosis (57.9%), while the women exhibited mildly or moderately eroded cortexes (66.5%) and a high risk of osteoporosis (43.1%). The MCW values show that the subjects were still considered in the normal category for both men and women (cutoff > 3.00 mm).

The OSTA index values for men, ranked from low risk to high risk, are provided in Table 2. Per QUS, the OSTA index significantly decreased in its SOS, BUA, and SI values. On the high-risk OSTA index, the z-score was in the osteopenia category. On the moderate- and high-risk OSTA indexes, the t-scores corresponded to osteopenia and osteoporosis.

The OSTA index values for women, ranked from low risk to high risk, also showed a significant decrease in BUA, SOS, and SI values (Table 3). While the z-score was still within normal values on the three OSTA indexes, all of the t-score values were in the osteopenia category. The panoramic images revealed that the MCW and PMI values in the cutoff range for high risk of osteoporosis demonstrated high risk of OSTA for both men and women. In addition, low BMI values (i.e., being underweight) revealed a significant relationship with the high-risk OSTA index in men and women.

The coefficient correlations based on the z-score and t-score values are shown in Table 4. Based on the z-scores, the highest-to-lowest correlation coefficient values were age, SI, BUA, SOS, OSTA, MCW, BMI, PMI, MCI, and the number of remaining teeth. Based on the t-scores, the highest-to-lowest correlation coefficient values were age, SI, BUA, SOS, OSTA, MCI, MCW, sex, PMI, BMI, and the number of remaining teeth. The MCI, MCW, and PMI had lower coefficient correlation values than the OSTA index, based on z-score (r = −0.563, *p* < 0.001) and t-scores (r = −0.740, *p* < 0.001). However, the OSTA correlation coefficient was still lower than the values of BUA, SOS, and SI shown on the QUS.

The OSTA index’s performance is shown in Table 5. The table also displays the sensitivity and specificity at the cutoff values of the OSTA index ≤−1 and t-score ≤ −1.0 shown via QUS. The results indicate that sensitivity was higher in women, but specificity was lower than in men. At the same time, the AUC results show that the OSTA values exhibited a significant difference, with a cutoff t-score of ≤−1.0 via QUS in men (*p* < 0.001, 95% CI 0.751–0.874) and women (*p* < 0.001, 95% CI 0.787–0.938) (Figure 3).

## 4. Discussion

This study found that the OSTA index had a significant relationship with MIC, MCW, and PMI in men, while only MCW and PMI showed correlation in women (Table 2 and Table 3). For subjects who were categorized as at either medium or high risk of developing osteoporosis, panoramic images revealed that the OSTA index showed significantly lower MCW values and PMI ratio values. For the MCW and PMI ratio, several studies have utilized threshold values of <3.00 and <0.3, respectively, to determine those who are at high risk of developing osteoporosis [15,16]. In this study, it was found that both men and women in the high-risk category of the OSTA index also had a high risk of osteoporosis based on their MCW values and PMI ratio values. Both men and women with a moderate or high risk of osteoporosis on the OSTA index had significantly lower BMD measurement values (SI, BUA, SOS, z-score, and t-score) than those with a low risk of osteoporosis on the OSTA index, as shown by the QUS tool. These results for SI, BUA, SOS, and t-scores are in accordance with results from other studies that used the QUS tool as a screening tool for osteoporosis [9]. In addition, increasing age in subjects had a significant relationship with the risk of osteoporosis in panoramic indices or the OSTA index in both men and women. Studies have shown that the decline in bone density continues with increasing age. In men, bone loss averages between 0.2% and 0.5% per year. In women, there is an accelerated rate of bone loss at menopause of about 1–2%, which increases to 3–5% during the first 5–8 years post-menopause. These results are in line with a study stating that with increasing age, the risk of developing osteoporosis will also increase [14]. This confirms that the OSTA index is a simple method that can be used in general dental practice to identify older patients at risk of osteoporosis.

The correlation coefficient value of the OSTA index was found to be higher than that of the three method indices for the panoramic images that applied z-scores and t-scores using the QUS tool (Table 4). It should be noted that the correlation coefficient values obtained for the OSTA index were based on the QUS tool and not DEXA. Many other studies using DEXA as a diagnostic test for osteoporosis have demonstrated the usefulness of the three panoramic indices for detecting patients at risk for osteoporosis [17,18,19]. Therefore, using the OSTA index and the three panoramic indices can provide more specific results in patients suspected of osteoporosis. In our opinion, dentists should educate patients about their risk of osteoporosis, which could impact the outcome of dental treatment, by using the cutoff value of the OSTA index category for high risk (OSTA < −4) with class 3 of the MCI values, as well as an MCW below 3 mm and a PMI ratio below 0.3 [20,21]. Furthermore, dentists can encourage older patients to undergo a DEXA diagnostic test if the patient is found to be at risk of osteoporosis so that they can receive appropriate treatment to prevent future fractures.

Although the primary purpose of this study was not designing a diagnostic test, the performance of the OSTA index needs to be evaluated according to QUS. The receiving–operating characteristic used a cutoff t-score of ≤−1.0 to identify patients between normal and osteopenia/osteoporosis as early as possible. The OSTA index cutoff for sensitivity and specificity was ≤−1 to differentiate low risk from moderate or high risk of osteoporosis. Sherchan et al. recommend a cutoff of ≤−1 for the OSTA index and a t-score of ≤−1 using QUS, citing these values as the most accurate [7]. Their study showed that the sensitivity and AUC values of the OSTA index were higher in women than in men, while specificity was lower in women than in men. Notably, there are not many studies that use male participants. This may be because body height in men influences QUS scores more than age and body weight [22]. One study that used male participants was by Zha et al., who conducted research using QUS with 472 elderly Chinese men. They noted sensitivity at 80.4%, specificity at 59.7%, and an AUC of 0.762 [23]; these results resemble those of the current study.

Regarding the results for women in the present study, they somewhat resemble the results of prior studies that used DEXA as a diagnostic tool. The OSTA index formula was first proposed by Koh et al. using DEXA as a diagnostic tool, in which they validated the OSTA index on 1123 Japanese women with a cutoff of −1 [6]. They obtained sensitivity at 98% and specificity at 29% [6]. While the differences in diagnostic test tools and cutoffs between the study by Koh et al. and the present one prevent valid comparisons, patterns of high sensitivity and low specificity among the female participants in both studies may nonetheless be observed. Meanwhile, Chin et al. obtained different results in their study of 283 men and 362 women [9]. The sensitivity, specificity, and AUC values in their study were 79.3%, 57.7%, and 0.685 in men and 54.8%, 69.3%, and 0.620 in women, respectively [9].

Differences in the use of the cutoff OSTA index, QUS t-score, number of subjects, and age categories between the present study and prior research show varying values for sensitivity, specificity, and AUC. However, the current study’s results nevertheless indicate that applying the OSTA index to identify patients at risk of osteoporosis would be useful in general dental practice.

The limitation of this study was the unavailability of data using DEXA as an osteoporosis diagnostic test tool. The participants’ age group, healthy conditions of the participants (meaning there were no indications for DEXA radiation examination), and a lack of DEXA examination facilities made QUS the preferred tool in this study. This study was conducted to evaluate the relationship between these variables. Thus, the performance of OSTA, the three panoramic indices, and QUS to identify osteoporosis cannot be compared with studies using DEXA. Further research using DEXA is needed, as this tool is the gold standard for diagnostic performance with all of the assessed variables.

## 5. Conclusions

The OSTA index is a simple method that can be used in general dental practice to identify patients suspected of having osteoporosis.

## Figures and Tables

**Figure 1 dentistry-11-00034-f001:**
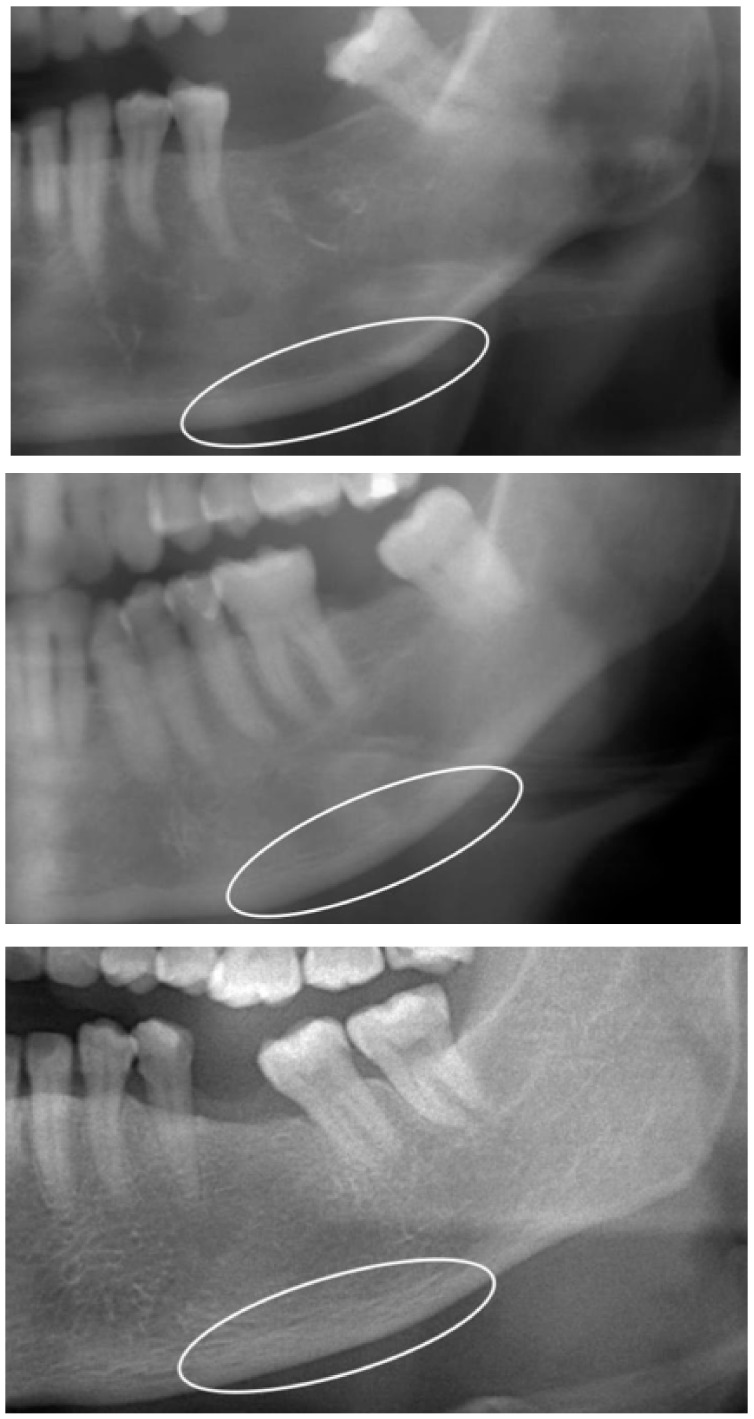
Mandibular cortical index (MCI). Participants were divided into three categories (C1–C3) according to the MCI. Class 1 (normal cortex): the endosteal margin of the cortex was even and sharp on both sides. Class 2 (mildly or moderately eroded cortex): the endosteal margin showed semilunar defects (lacunar resorption) or seemed to form endosteal cortical residues (one to three layers) on one or both sides. Class 3 (severely eroded cortex): the cortical layer formed healthy endosteal cortical residues and was clearly porous. If the MCIs of the right and left sides were different, then the side with the most severe MCI classification was chosen.

**Figure 2 dentistry-11-00034-f002:**
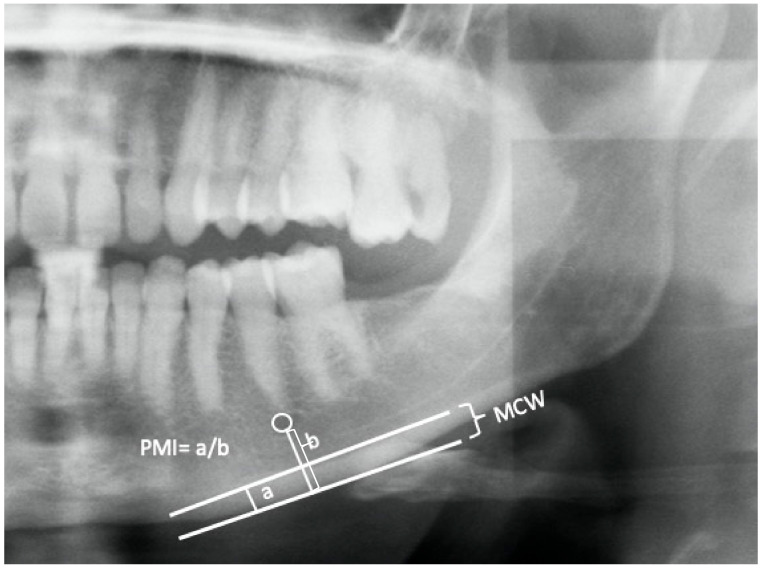
Mandibular cortical width (MCW) and panoramic mandibular index (PMI). The mandibular cortical width (MCW) is the distance between the inferior and superior cortex. The panoramic mandibular index (PMI) ratio is the ratio of the measurements of (**a**) the cortical thickness and (**b**) the point from the inferior border of the mental foramen (white circle) to the inferior border of the mandible.

**Figure 3 dentistry-11-00034-f003:**
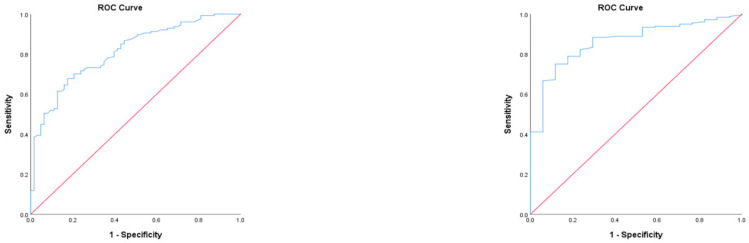
The receiver operating characteristics (ROC) curves for the Osteoporosis Self-Assessment Tool for Asians (OSTA) values in men and women. There were significant differences between the OSTA values and cutoff of t-score ≤ −1.0 on quantitative ultrasound (QUS) (*p* < 0.001) in men and women. The areas under the curve in men and women were 0.812 (95% confidence interval 0.751–0.874) and 0.862 (95% confidence interval 0.787–0.938), respectively.

**Table 1 dentistry-11-00034-t001:** Demographic data of the 387 participants.

Variable	Men (*n* = 190)		Women (*n* = 197)		*p*-Value
	Mean ± SD	Median (min–max)	Mean ± SD	Median (min–max)	
Age (in years)	58.99 ± 6.7	58 (50–70)	61.97 ± 4.9	63 (50–70)	<0.001 *
Quantitative ultrasound (QUS)					
Speed of sound (SOS) (m/s)	1537.41 ± 21.5	1534 (1486–1601)	1518.36 ± 20.8	1515 (1489–1707)	<0.001 *
Calcaneus broadband attenuation of sound (BUA) (dB MHz)	104.59 ± 11.6	103.6 (70.2–139.3)	91.24 ± 7.9	90.4 (66.7–124)	<0.001 *
Stiffness index (%)	80.55 ± 13.1	78.83 (46.5–113.8)	66.27 ± 9.1	64.79 (45.8–107.7)	<0.001 *
z-score	−0.03 ± 1.2	−0.16 (−3–3.3)	0.04 ± 0.8	−0.07 (−2.2–3.4)	not significant
t-score	−1.39 ± 1.1	−1.5 (−4.1–1.7)	−1.99 ± 0.7	−2.08 (−3.9–1)	<0.001 *
Panoramic radiography					
Mandibular cortical index (MCI)					
Class 1	134 (70.6%)		34 (17.3%)		<0.001 **
Class 2	47 (24.7%)		131 (66.5%)		
Class 3	9 (4.7%)		32 (16.2%)		
Mandibular cortical width (MCW) (mm)	3.22 ± 0.6	3.32 (1.3–4.3)	3.11 ± 0.6	3.17 (1.7–4.6)	<0.001 *
Panoramic mandibular index (PMI)	0.28 ± 0.05	0.29 (0.12–0.37)	0.27 ± 0.05	0.28 (0.12–0.37)	not significant
Osteoporosis Self-Assessment Tool for Asians (OSTA):					
Low risk (OSTA > −1)	110 (57.9%)		40 (20.3%)		<0.001 **
Medium risk (−4 < OSTA < −1)	61 (32.1%)		72 (36.6%)		
High risk (OSTA <−4)	19 (10%)		85 (43.1%)		
Body mass index (kg/m^2^)	21.85 ± 2.8	21.96 (14.6–28.5)	22.16 ± 3.37	21.98 (15–31.8)	not significant
Number of remaining teeth	15.41 ± 10	16.5 (0–32)	15.75 ± 9.3	17 (0–32)	not significant

* Independent-samples Mann-Whitney U Test. ** Chi-square test.

**Table 2 dentistry-11-00034-t002:** The relationships of the Osteoporosis Self-Assessment Tool for Asians (OSTA) with all of the variables in men.

Variable	Low Risk (OSTA > −1)		Medium Risk (−4 < OSTA < −1)		High Risk (OSTA < −4)		*p*-Value
	Mean ± SD	Median (min–max)	Mean ± SD	Median (min–max)	Mean ± SD	Median (min–max)	
Age (in years)	54.9 ± 4.4	54 (50–67)	63.41 ± 5.1	65 (52–70)	68.53 ± 2.6	69 (61–70)	<0.001 *
Quantitative ultrasound (QUS)							
Speed of sound (SOS) (m/s)	1545.69 ± 18.9	1542 (1510–1593)	1529.28 ± 19.8	1526 (1486–1601)	1515.53 ± 15.1	1515 (149–1543)	<0.001 *
Calcaneus broadband attenuation of sound (BUA) (dB MHz)	110.5 ± 9.2	108.8 (96–139.3)	98.76 ± 8.4	98.5 (75.2–122.2)	89.12 ± 9.6	90.2 (70.2–110)	<0.001 **
Stiffness index (%)	86.83 ± 10.7	84.58 (69.1–113.8)	74.37 ± 10.6	73.2 (46.5–107.9)	64.05 ± 9.4	63.1 (52.6–85.7)	<0.001 *
z-score	0.56 ± 0.9	0.37 (−0.9–3.3)	−0.6 ± 0.9	−0.69 (−2.8–2)	−1.5 ± 0.8	−1.5 (−3–0.4)	<0.001 **
t-score	−0.85 ± 0.9	−1.02 (−2.2–1.7)	−1.9 ± 0.8	−1.99 (−4–0.5)	−2.77 ± 0.8	−2.79 (−4.1– −0.9)	<0.001 **
Panoramic radiography							
Mandibular cortical index (MCI)							
Class 1	86 (78.2%)		38 (62.3%)		10 (52.6%)		<0.005 ***
Class 2	23 (20.9%)		19 (31.1%)		5 (26.3%)		
Class 3	1 (0.9%)		4 (6.6%)		4 (21.1%)		
Mandibular cortical width (MCW) (mm)	3.36 ± 0.4	3.38 (1.6–4.3)	3.11 ± 0.7	3.2 (1.6–4.3)	2.72 ± 0.7	2.85 (1.3–3.8)	<0.005 *
Panoramic mandibular index (PMI)	0.296 ± 0.03	0.29 (0.22–0.37)	0.266 ± 0.05	0.275 (0.17–0.35)	0.253 ± 0.06	0.25 (0.12–0.36)	<0.05 *
Body mass index (kg/m^2^)	23.15 ± 2.1	23.1 (18.7–28.5)	20.46 ± 2.2	20.8 (15.7–15.2)	18.79 ± 3.1	18.6 (14.6–25.9)	<0.001 *
No. of remaining teeth	16.64 ± 9.8	20 (0–32)	14.15 ± 9.8	14 (0–30)	12.32 ± 10.6	8 (0–31)	Not significant

* Kruskal–Wallis Test. ** One-way ANOVA test. *** Fisher’s exact test.

**Table 3 dentistry-11-00034-t003:** The relationship of the Osteoporosis Self-Assessment Tool for Asians (OSTA) with all of the variables in women.

Variable	Low Risk (OSTA > −1)		Medium Risk (−4 < OSTA < −1)		High Risk (OSTA < −4)		*p*-Value
	Mean ± SD	Median (min–max)	Mean ± SD	Median (min–max)	Mean ± SD	Median (min–max)	
Age (in years)	55.75 ± 3.86	55 (50–63)	61.14 ± 3.6	66 (59–70)	65.61 ± 2.6	66 (59–70)	<0.001 *
Quantitative ultrasound (QUS)							
Speed of sound (SOS) (m/s)	1529.48 ± 20.3	1527 (1502–1592)	1521.38 ± 25.4	1518 (1492–1707)	1510.56± 12.1	1510 (1489–1556)	<0.001 *
Calcaneus broadband attenuation of sound (BUA) (dB MHz)	100 ± 7.3	99.1 (87.1–124)	91.93 ± 6,2	91.8 (72.6–109.8)	86.49 ± 5.2	86.8 (66.7–106.4)	<0.001 *
Stiffness index (%)	86.83 ± 10.7	84.58 (69.1–113.8)	74.37 ± 10.6	73.2 (46.5–107.9)	64.05 ± 9.4	63.1 (52.6–85.7)	<0.001 *
z-score	0.95 ± 0.8	0.79 (−0.09–3.4)	0.13 ± 0.6	0.07 (−1.6–1.8)	−0.46 ± 0.5	−0.45 (−2.2–1.8)	<0.001 *
t-score	−1.19 ± 0.7	−1.33 (−2.1–1.02)	−1.91 ± 0.5	−1.96 (−3.4–−0.4)	−2.43 ± 0.8	−2.43 (−3.9–−0.4)	<0.001 *
Panoramic radiography							
Mandibular cortical index (MCI)							Not significant
Class 1	7 (17.5%)		14 (19.4%)		13 (15.3%)		
Class 2	29 (72.5%)		49 (68.1%)		53 (62.4%)		
Class 3	4 (10%)		9 (12.5%)		19 (22.4%)		
Mandibular cortical width (MCW) (mm)	3.29 ± 0.5	3.21 (2.03–4.3)	3.19 ± 0.5	3.27 (2.03–4.3)	2.94 ± 0.6	2.97 (1.7–4.6)	<0.005 **
Panoramic mandibular index (PMI)	0.3 ± 0.04	0.3 (0.18–0.37)	0.272 ± 0.058	0.28 (0.12–0.37)	0.267± 0.057	0.27 (0.12–0.36)	<0.05 *
Body mass index (kg/m^2^)	25.5 ± 2.6	23.35 (21–31.77)	23.05 ± 2.5	23 (18.1–30.6)	19.8 ± 2.6	19.7 (15–27.9)	<0.001 **
Number of remaining teeth	15.53 ± 9.7	16 (0–30)	15.57 ± 8.9	17 (0–29)	16.01 ± 9.6	17 (0–29)	Not significant

* Kruskal–Wallis test. ** One-way ANOVA test.

**Table 4 dentistry-11-00034-t004:** Correlations of all variables based on z-score and t-score according to Spearman’s rho correlation coefficient value.

Variable	z-Score		t-Score	
	Correlation Coefficient (r)	*p*-Value	Correlation Coefficient (r)	*p*-Value
Age (in years)	−0.774	<0.001	−0.895	<0.001
Sex (1: men; 2: women)	0.57	Not significant	−0.339	<0.001
Quantitative ultrasound (QUS)				
Speed of sound (SOS) (m/s)	0.654	<0.001	0.827	<0.001
Calcaneus broadband attenuation of sound (BUA)(dB MHz)	0.709	<0.001	0.934	<0.001
Stiffness index (%)	0.726	<0.001	0.938	<0.001
Panoramic radiography				
Madibular cortical index (MCI) (1: class 1; 2: class 2; 3: class 3)	−0.211	<0.001	−0.413	<0.001
Mandibular cortical width (MCW) (mm)	0.326	<0.001	0.376	<0.001
Panoramic mandibular index (PMI)	0.286	<0.001	0.308	<0.001
Osteoporosis Self-Assessment Tool for Asians (OSTA) (1: low risk; 2: medium risk; 3: high risk)	−0.563	<0.001	−0.740	<0.001
Body mass index (kg/m^2^)	0.301	<0.001	0.270	<0.001
Number of remaining teeth	0.116	<0.05	0.102	<0.05

**Table 5 dentistry-11-00034-t005:** The sensitivity and specificity at cutoff values of OSTA index ≤−1 and t-score ≤ −1.0 on QUS.

OSTA Index	Normal (%)	Osteopenia and Osteoporosis (%)	Total	Sensitivity	Specificity
Men					
OSTA index > −1	55 (50)	55 (50)	110 (100%)	90%	50%
OSTA index ≤ −1	8 (10)	72 (90)	80 (100%)		
Women					
OSTA index > −1	12 (30)	28(70)	40 (100%)	96.8%	30%
OSTA index ≤ −1	5 (3.2)	152 (96.8)	157 (100%)		
Men & women					
OSTA index > −1	67 (44.7)	83 (55.3)	150 (100%)	94.5%	44.7%
OSTA index ≤ −1	13 (5.5)	224 (94.5)	237 (100%)		

## Data Availability

The data that support the findings of this study are available on request from the corresponding author, [B.K]. The data are not publicly available due to privacy or ethical restrictions.

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
