# Peer review of "Correlations of the Osteoporosis Self-Assessment Tool for Asians (OSTA) and Three Panoramic Indices Using Quantitative Ultrasound (QUS) Bone Densitometry"

_dentistry, 2023, doi:10.3390/dj11020034_

Round 1

Reviewer 1 Report

Overall, this is a very well planned and written study.  The results are interesting.

Abstract: The abstract is well written and easy to understand.

Keywords: simple for readers to understand and are relevant to the content.

Introduction: The introduction provides a good, generalized background of the topic. However, to make the introduction more substantiated, the authors could make the following improvements:

It reads “ A high risk of osteoporosis can affect patients’ oral health and treatment in dentistry 32 if the condition is not detected early.” Please remove “A high risk” 

It reads “Body weight, patient age, and panoramic radiography are typically measured during dentistry appointments, especially for older patients who want to make prostheses” Please add a reference. Not in all countries of the world, dentists typically determine the patient's weight. 

In the introduction, it should be explained in more detail why it is important to compare OSTA with the QUS T-score and not with DXA T-sore, which is the gold standard for the diagnosis of osteoporosis. 

It reads “Moreover, the relationship of the value of bone mineral density (stiffness index, or SI), the speed of sound (SOS), and the Calcaneus broadband attenuation of sound (BUA) with the QUS tool also need to be investigated.” Please explain in more detail why this is important.

Material and Methods:

This study has considered every important point required. Ethical consideration and inclusion-exclusion criteria have been explained properly.

However, authors must describe in more detail how patients were recruited for the study (or patients of a specific clinic, etc.).

Describe how the number of remaining teeth was determined.

However, it should be specified how many experts performed the measurements and how many and at what intervals the measurements were repeated.

Please clarify how the patient weight and height data were obtained.

Please provide a power analysis for your final sample size and group breakdown.

Results: well-written results and tables are self-explanatory. Data are well presented, and no need for any supplementary figures or tables.

Discussion: The discussion is well written.

However, it is necessary to describe more the effect of age on panoramic indices and the risk of osteoporosis. Especially considering that there was a statistically significant difference between OSTA risk groups by age.

Conclusions: It readsCombining the OSTA index and the three panoramic indices can increase the probability of  identifying patients suspected of having osteoporosis.This statement is based more on the assumption of the authors than the results of this study.

Author Response

The answer to reviewer comments.

Thank you for helping us improve our research.

Our point-by-point reply to reviewer comments (All changes are shown in red):

Introduction

Point 1: “A high risk of osteoporosis can affect patients’ oral health and treatment in dentistry 32 if the condition is not detected early." Please remove "A high risk."

Response 1: Thank you for the advice. We decided to delete the words "a high risk." As we have deleted "a high risk" from the introduction, we also have deleted it in the abstract.

Osteoporosis can affect patients’ oral health and treatment in dentistry if the condition is not detected early.

Point 2: “Body weight, patient age, and panoramic radiography are typically measured during dentistry appointments, especially for older patients who want to make prostheses” Please add a reference. Not in all countries of the world, dentists typically determine the patient's weight. 

Response 2: Thank you for the comments. We understand the reviewer's concern. We have added a reference in the sentences. The reference used comes from Japan which measures patients' weight related to dental prosthetics following the contents in the sentences. Hopefully, they can meet the reviewer's criteria.

Body weight, patient age, and panoramic radiography are typically measured during dentistry appointments, especially for older patients who want to make prostheses [5].

5. Kusama, T.; Nakazawa, N.; Kiuchi, S.; Kondo, K.; Osaka, K.; Aida, J. Dental prosthetic treatment reduced the risk of weight loss among older adults with tooth loss. J Am Geriatr Soc. 2021, 69, 2498–2506

Point 3: In the introduction, it should be explained in more detail why it is important to compare OSTA with the QUS T-score and not with the DXA T-score, which is the gold standard for diagnosing osteoporosis.

Response 3: Thank you for the comments, as we have already stated the reasons for the limitation of this study. We also added an explanation and a reference in the introduction.  

Considering the indications for DXA examination are all women over 65 years and men over 70 years or aged 50 years ages who have experienced a fracture with minimal or no trauma, excluding pathologic fractures [13]. Therefore, the DXA examination does not apply to this study.

Reference:

13. Krugh, M.; Langaker, M.D. Dual Energy X-ray Absorptiometry. In: StatPearls [Internet]. Treasure Island (FL): StatPearls Publishing, 2022. Available from: https://www.ncbi.nlm.nih.gov/books/NBK519042/

Point 4: It reads “Moreover, the relationship of the value of bone mineral density (stiffness index, or SI), the speed of sound (SOS), and the Calcaneus broadband attenuation of sound (BUA) with the QUS tool also need to be investigated.” Please explain in more detail why this is important.

Response 4: Thank you for the comments. We have added a sentence and a reference to improve the descriptive details in the introduction.

These parameters are usually used as research variables for measuring bone densitometry besides the Z-score and T-score [9].

Reference:

  1. Chin, K.Y.; Low, N.Y.; Kamaruddin, A.A.A.; Dewiputri, W.I.; Soelaiman, I.N. Agreement between calcaneal quantitative ultrasound and osteoporosis self-assessment tool for Asians in identifying individuals at risk of osteoporosis. Ther. Clin. Risk Manag. 2017, 13, 1333–1341.

Material and methods:

Point 1: authors must describe in more detail how patients were recruited for the study (or patients of a specific clinic, etc.).

Response 1: Thank you for remembering us. We added the explanation.

Research with a cross-sectional design was carried out initially on 400 healthy older people (aged 50 to 70 years) from community health centers in the province of West Java, Indonesia, in 2019.

Point 2: Describe how the number of remaining teeth was determined. Please clarify how the patient weight and height data were obtained.

Response 2: Thank you for improving our manuscript. We added the information on the material and methods.

The patient’s weight and height were measured with the same instrument using a medical mechanical body weighing scale (SH-8024, Jiangsu Suhong Medical Device Co. Ltd., China). Weight is measured in kg, while height is measured in cm. Body mass index (BMI) is obtained by calculating the weight (kg) divided by the square of the height (m). The number of remaining teeth was analyzed until third molar on panoramic radiographs with the following criteria: healthy, caries, or restored. Teeth indicated for extraction and remaining roots were not included.

Point 3: However, it should be specified how many experts performed the measurements and how many and at what intervals the measurements were repeated.

Response 3: Thank you for remembering us. We have added the sentences.

Intra-and inter-observer agreement measurements were performed by the two radiologists on 50 randomly selected panoramic views with a weekday interval between measurements.

Point 3: Please provide a power analysis for your final sample size and group breakdown.

Response 3: Thank you for the comments. The power analysis was set at 0.8, and all participants were divided into groups by sex. We added these sentences to the statistical analysis.

Discussion

Point 1: However, it is necessary to describe more the effect of age on panoramic indices and the risk of osteoporosis. Especially considering that there was a statistically significant difference between OSTA risk groups by age.

Response 1: Thank you for the advice. We added the sentences in the discussion.

In addition, increasing age in subjects has a significant relationship with the risk of osteoporosis through panoramic indices or OSTA index, both in men and women. Studies have shown that the decline in bone density continues with increasing age. In men, bone loss averages between 0.2 and 0.5% per year. In women, there is an accelerated rate of bone loss at menopause of about 1-2% and increases to 3-5% during the first 5-8 years of post-menopause. These results are in line with research which states that with increasing age, the risk of developing osteoporosis will also increase [14].

14. Who are candidates for prevention and treatment for osteoporosis? Osteoporos. Int. 1997, 7, 1–6.

Point 1: It reads, " Combining the OSTA index and the three panoramic indices can increase the probability of  identifying patients suspected of having osteoporosis.” This statement is based more on the assumption of the authors than the results of this study.

Response 2: Thank you for the advice. We have modified the sentence.

The OSTA index is a simple method that can be used in general dental practice to identify patients suspected of having osteoporosis.

Reviewer 2 Report

This paper studies the relation between osteoporosis and 3 panoramic indices. This work is very interesting. However, I have some questions:

1) MCI is explained in the first paragraph of page 4. But, is not clear that you are talking about MCI.

2) In Figure 2, how b is measured? Is the maximum distance, the minimum distance? Because, if another teeth is chosen another b value is obtained. That is not clear.

3) The relation between OSTA and MCI, MCW and PMI is not clear for me. For instance, in women PMI is equal for the medium risk and high-risk groups. For men is better.

4) Table 5 is correct? Index >-1, so less risk, more osteoporosis, compared with index <-1?

5) Dexa results are not possible in this study. But can be interesting to compare QUS with MCI, MCW and PMI. Can you include that?

Some minor issues:

1) in line 105 please delete > and write “more than”.

Author Response

The answer to reviewer 2 comment.

Thank you for helping us improve our research.

Our point-by-point reply to reviewer comments (All changes are shown in red):

Introduction

Point 1: MCI is explained in the first paragraph of page 4. But, is not clear that you are talking about MCI. 

Response 1: Thank you for the question. We explained more detail how to do the measurement of the MCI.

All participants were divided into three categories (C1-C3) according to the criteria; Class 1 (normal cortex): the endosteal margin of the cortex was even and sharp on both sides. Class 2 (mildly or moderately eroded cortex): the endosteal margin showed semilunar defects (lacunar resorption) or seems to form endosteal cortical residues (one to three layers) on one or both sides. Class 3 (severely eroded cortex): the cortical layer formed healthy endosteal cortical residues and was clearly porous. If the MCI between the right and left side is different, then the class chosen is the one with the most severe MCI.

Point 2: In Figure 2, how b is measured? Is the maximum distance, the minimum distance? Because, if another teeth is chosen another b value is obtained. That is not clear. 

Response 2: Thank you for the question. The anatomical landmark used is the mental foramen which is marked with a white circle. The point b is measured from point from the inferior border of the mental foramen (white circle) to the inferior border of the mandible. We add text and put lines as marks in the figure to make it more clear.

Point 3: The relation between OSTA and MCI, MCW and PMI is not clear for me. For instance, in women PMI is equal for the medium risk and high-risk groups. For men is better. 

Response 3: Thank you for the comment. We checked again in SPSS, it turns out that the results in women are from rounding up to two decimal places where the actual number medium risk of OSTA is 0.2716 ± 0.05781 becomes 0.27 ± 0.06, while for high risk is 0.2675 ± 0.05720 becomes 0.27 ± 0.06. To avoid misunderstanding, we use rounding up to three decimal places for PMI only. The medium risk becomes 0.272 ± 0.058, whereas for the high risk becomes 0.267 ± 0.057. We have change the number in the table.

Point 4: Table 5 is correct? Index >-1, so less risk, more osteoporosis, compared with index <-1?

Response 4: Thank you for the correction. We were not aware of the typo that occurred in the table 5. We are very grateful for the revision. An error occurred in the symbol used (“OSTA index >-1” and “OSTA index ≤-1”). But the values from the existing table are correct. We have made the revision in the table 5.

Point 5: Dexa results are not possible in this study. But can be interesting to compare QUS with MCI, MCW and PMI. Can you include that? 

Response 5: Thank you for the advice. The relationship between QUS and three panoramic indices (MCI, MCW, and PMI) has been carried out in the previous study. Thank you again for the idea. Herewith we provide the study which discusses the relationship between QUS and MCI, MCW, PMI.

Kiswanjaya, B.; Priaminiarti, M.; Bachtiar-Iskandar, H.H. Three panoramic indices for identification of healthy older people at a high risk of osteoporosis. Saudi Dent. J. 2022, 34, 503–508

Some minor issues:

Point 6: in line 105 please delete > and write “more than”.

Response 6: Thank you for the comment. We have changed it.

MCI, MCW, and PMI values were measured by specialists with more than 10 years of expertise in oral radiology.

Round 2

Reviewer 2 Report

Thanks for the kind answer. In my opinion this version can be published.

Author Response

Point 1: Thanks for the kind answer. In my opinion this version can be published.

Response: 

Thank you very much. We also would like to express our gratitude for having reviewed and provided many significant comments to improve our manuscript.